# Comparison of new and old BacT/ALERT aerobic bottles for detection of *Candida* species

**Kye Won Choe[ID], Yong Kwan Lim[ID], Mi-Kyung Lee[ID]***

Department of Laboratory Medicine, Chung-Ang University College of Medicine, Seoul, Republic of Korea

* cpworld@cau.ac.kr

**Data Availability Statement:** All relevant data are within the manuscript and its Supporting Information files.

**Funding:** This work was supported by the National Research Foundation of Korea (NRF) grant funded

## Abstract

### Purpose

A new version of aerobic blood culture media has been developed for the BacT/ALERT (bioMérieux) blood culture system. We evaluated the time to detection and yeast cell counts in positive blood cultures for each *Candida* spp. according to changes in media.

### Methods

Isolates from defibrinated horse blood were inoculated into three types of bottles: the old version of aerobic bottle, new version of aerobic bottle, and anaerobic bottle. All bottles were incubated in the BacT/ALERT Virtuo blood culture system. The time to detection was monitored for each bottle, and yeast cell counts were performed immediately after testing positive, determined via the plate count method. Clinical retrospective data of the candidemia samples before and after aerobic bottle change also were analyzed.

### Results

The median time to detection was 52.47 hours in the old aerobic bottles versus 19.92 hours in the new aerobic bottles ($P < 0.001$) for *Candida glabrata*, and standard and clinical strains showed similar results. *C. albicans* (27.6 to 24.95 hours) and *C. guilliermondii* (28.92 to 26.9 hours) had shorter time to detection. However, *C. auris* (25.43 to 28.25 hours) had a longer time to detection in the new aerobic bottle. The retrospective clinical analysis showed a significant decrease in time to detection (45.0 to 19.4 hours) for *C. glabrata*, which is consistent with our simulated study result for *C. glabrata*. As a result of analysis including all blood specimens, *C. tropicalis* showed a significant delay in time to detection in new aerobic bottles. In an analysis limited to peripheral blood specimens, the time to detection of *C. parapsilosis* was longer in new aerobic bottles than in old aerobic bottles.

### Conclusion

Most *Candida* species did not show remarkable TTD differences, but TTD of *C. glabrata* was markedly reduced in the New FA Plus bottle. The reduction of time to detection enables faster detection and therapeutic approach for *C. glabrata* infections.

by the Korean government (MSIT) (No.2020R1A5A1018052). The funders had no role in the study design, data collection and analysis, decision to publish, or manuscript preparation.

**Competing interests:** The authors have declared that no competing interests exist

## Introduction

Candidemia is an increasingly major healthcare-associated fungal infection associated with high morbidity and mortality [1, 2]. The epidemiology of candidemia varies according to the geographical region, period, and population; however, recent studies have reported a global shift towards other *Candida* spp. from *Candida albicans*, although *C. albicans* is still the predominant species [2, 3].

*C. glabrata* (also named as *Nakaseomyces glabratus*) is a common pathogen whose proportions are increasing among non-*C. albicans* infections and is causing bloodstream infections worldwide, including South Korea [4–6]. Candidemia caused by *C. glabrata* has clinical significance due to antifungal resistances, including intrinsic decreases in susceptibility to azole agents and resistances to amphotericin B [6]. It has been reported that time to detection (TTD) of *C. glabrata* was more affected by the blood culture system and medium than *C. albicans* [7, 8].

The new version of aerobic blood culture media has been developed for the BacT/ALERT (bioMérieux) blood culture system. Although the exact formulation of the bottle media is proprietary and not disclosed, it was reported that trace elements were added and several components unrelated to microbial recovery were removed. The primary purpose of the formulation changes was to increase the growth performance of several microorganisms particularly *C. glabrata*. The new BacT/ALERT aerobic media (New FA Plus) has been shown to have equivalent performance compared with the old version, except for the reduced TTD of some microorganisms.

There have been no studies on the effects that the change in aerobic blood culture bottle has on the TTD and growth performance of various *Candida* spp. Therefore, we aimed to analyze the changes in TTD and yeast cell count for several *Candida* spp. according to media changes through a simulated blood spiking study and clinical retrospective data analysis.

## Materials and method

### Study design

The study was performed at Chung-Ang University Medical Center laboratory in Seoul, South Korea. The study was conducted in two ways: one is an experiment to check TTD and yeast cell count by inoculating isolates into horse blood, and the other is a retrospective comparative analysis of TTD in blood culture data obtained from outpatient and hospitalized patients.

### Preparing a simulated blood culture study

The *Candida* isolates evaluated in this study (n = 14) are listed in Table 1. These isolates included nine strains (American Type Culture Collection, ATCC; National Culture Collection for Pathogens of Korea, NCCP) and five clinical isolates from the blood of patients at Chung-Ang University Hospital who were diagnosed with candidemia. All the isolates were received and analyzed by the Chung-Ang University Medical Center laboratory. This study was approved by the IRB at the Chung-Ang University Hospital (IRB No.2307-018-19480). The IRB waived the need for informed consent and additional medical ethics review for this study, because the clinical isolates were properly de-identified and anonymized, according to institutional policy. Isolates were stored as 50% glycerol stocks at -70˚C. Before testing, the isolates were subcultured on Sabouraud dextrose agar (SDA) plates at 35˚C. After 24 to 48 hours of

**Table 1. *Candida* spp. tested under the simulation study.**

| *Candida* species | Isolate strain designations[a] |
|---|---|
| *C. albicans* | ATCC 14053 |
| *C. auris* | NCCP 32684 |
| *C. dubliniensis* | ATCC MYA-646 |
| *C. guilliermondii* | ATCC 6260 |
| *C. krusei* | ATCC 6258 |
| *C. lusitaniae* | ATCC 34449 |
| *C. parapsilosis* | ATCC 22019 |
| *C. tropicalis* | ATCC 750 |
| *C. glabrata* | ATCC 2950 |
| | CL1 |
| | CL2 |
| | CL3 |
| | CL4 |
| | CL5 |

[a] ATCC, American Type Culture Collection; NCCP, National Culture Collection for Pathogens of Korea; CL, Clinical isolates.

incubation, 0.5 McFarland standard suspensions, which contain $1–5×10^6$ CFU/mL depending on the *Candida* spp. were prepared [9], and serial dilutions were performed. Approximately 125 CFU of each isolate, in 0.5% sterile saline, was inoculated in 8 mL of defibrinated horse blood and was incubated in three types of blood culture bottles: the old version of the aerobic culture bottle (Old FA Plus), New FA Plus, and the anaerobic culture bottle (FN Plus). Old FA Plus include 30 ml of complex medium and 1.6 g of adsorbent polymeric beads. New FA Plus has similar formulation except addition of essential elements and removal of trace elements. FN Plus is composed of 40 ml of complex medium and 1.6 g of adsorbent polymeric beads. The isolates were inoculated in triplicate. All the bottles were incubated in the BacT/ALERT Virtuo blood culture system until positivity was detected or for a maximum of five days.

## Time to detection

TTD is defined as the time from the start of the incubation to when the blood culture vial tests positive. The TTD was monitored for each bottle, and yeast cell counts were performed immediately after testing positive via the plate count method using SDA as the culture medium. The blood culture medium was also subcultured on blood agar plates to exclude contamination and to confirm the true-positive or true-negative detection results.

## Yeast cell count

The positive blood culture was mixed and serially diluted in saline to dilutions of $10^{-3}$, $10^{-4}$, and $10^{-5}$, the dilutions were vortexed, and 100 uL of each dilution was spread onto a separate SDA plate. After overnight incubation, the plates were examined for colonies. The number of CFU per milliliter was calculated based on the dilution factor [10].

## Retrospective data analysis for clinical blood culture results

We performed blood culture positive data analysis on the *Candida* spp. to confirm whether the data verified through the procedures using horse blood can be applied to clinical

candidemia cases between 1 September 2020 and 15 June 2022. We divided the data into two periods: before changing to the New FA Plus bottles (1 September 2020 to 31 May 2021) and after changing to the New FA Plus bottles (1 June 2021 to 15 June 2022). During the above periods, only *C. albicans, C. glabrata, C. parapsilosis, and C. tropicalis* were detected; therefore, the four species were included in the final analysis. Blood culture bottles with polymicrobial growth were excluded from the analysis since the polymicrobial culture could have influenced the TTD.

### Statistical analysis

Statistical analysis was performed to compare the TTD between the blood culture bottles using Mann-Whitney U test and Kruskal-Wallis test. Differences with *p*-values of greater than 0.05 were regarded as statistically significant. Datasets were entered into Microsoft Excel (Microsoft, Washington, USA) and analyzed using R version 4.1.0 (R Foundation for Statistical Computing, Vienna, Austria).

## Results

### Simulated study data

The hematocrit of the defibrinated horse blood was 41.4%. The TTD values were compared among the 14 isolates performed in triplicate for each bottle type (Table 2). The median TTD of *C. glabrata* was 52.47 hours using the Old FA Plus bottles versus 19.92 hours using the New FA Plus bottles ($P < 0.001$). The overall and clinical samples of *C. glabrata* showed similar results. *C. albicans* (27.6 versus 24.95 hours; $P = 0.2$) and *C. guilliermondii* (28.9 versus 26.9 hours; $P = 0.1$) also showed relatively shorter TTDs between the Old and New FA Plus bottles, however, the difference was not statistically significant. On the contrary, the TTD was longer for *C. auris* (25.43 versus 28.25 hours; $P = 0.0765$) in the New FA Plus bottle compared with the Old FA Plus bottle, although statistical significance of the difference could not be clarified.

The number of yeast in the bottles spanned a 3.78-log range, from $3.80 \times 10^3$ to $2.31 \times 10^7$ CFU/mL. The median yeast cell count for *C. glabrata* in the New FA bottle was approximately 2.2-fold higher than in the Old FA bottle ($P < 0.001$). Considering that the TTD of *C. glabrata* in the New FA Plus bottle was shorter than in the Old FA Plus bottle by 20 hours, the yeast cell count confirmed that the growth rate of the yeast was increased in the new bottle. In addition, although there was no significant difference, median yeast cell counts of New FA Plus was higher than Old FA Plus for *C. auris, C. parapsilosis,* and *C. tropicalis.*

### Aerobic versus anaerobic bottle

Although most *Candida* strains had no growth in the FN Plus bottles, *C. dubliniensis, C. lusitaniae,* and *C. glabrata* grew in two or more bottles. There was no significant difference in the TTD of *C. dubliniensis,* but there was an increase in TTD of *C. lusitaniae* compared with the FA bottles (Old FA Plus 18.6 hours and New FA Plus 16.78 hours versus FN Plus 23.52 hours; $P = 0.044$). The yeast cell count showed no significant difference between the aerobic and anaerobic bottles for *C. dubliniensis* and *C. lusitaniae.*

The TTD of *C. glabrata* in the FN Plus bottle was shorter than that of the Old and New FA Plus bottles (Old FA Plus 52.47 hours and New FA Plus 19.92 hours versus FN Plus 16.4 hours; $P < 0.001$). The increased proliferation of *C. glabrata* in the anaerobic bottle was consistent with several previous studies [11–13]. The yeast cell count was significantly higher in the FN Plus bottle compared with the Old FA Plus ($P<0.001$), but yeast cell count difference was not statistically significant with New FA Plus ($P = 0.1581$).

**Table 2. Time to detection and yeast cell count for *Candida* spp. in the simulated study.**

| | Strain[b] | Time to detection (hr) Median (25th percentile, 75th percentile) | | | P value[a] | Yeast cell count (CFU/mL) Median (25th percentile, 75th percentile) | | | P value[a] |
|---|---|---|---|---|---|---|---|---|---|
| | | Old | New | FN | | Old | New | FN | |
| *C. albicans* | ATCC 14053 | 27.6 (27.5, 27.6) | 24.95 (24.7, 25.1) | 29.6* | 0.2 | 2.05 (1.58, 2.53)×10³ | 11.8 (9.75, 12.3)×10⁴ | 500* | 0.2 |
| *C. auris* | NCCP 32684 | 25.43 (25.36, 25.77) | 28.25 (28.09, 28.25) | 17.1* | 0.08 | 3.2 (3.05, 3.20)×10⁶ | 7.20 (7.20, 7.85)×10⁶ | 1.40×10⁴* | 0.07 |
| *C. dubliniensis* | ATCC MYA-646 | 26.13 (25.80, 26.89) | 25.30 (24.81, 25.80) | 27.0(26.1, 26.99) | 0.2 | 2.8 (1.8, 4.15)×10⁶ | 2.2 (1.75, 3.60)×10⁶ | 2.30 (2.00, 5.10)×10⁶ | 0.2 |
| *C. guilliermondii* | ATCC 6260 | 28.92 (28.69, 29.00) | 26.9(26.85, 27.50) | - | 0.1 | 1.9 (1.75, 3.0)×10⁶ | 1.22 (1.12, 1.86)×10⁶ | - | 0.4 |
| *C. krusei* | ATCC 6258 | 21.14 (19,65, 22.64) | 17.68 (17.66, 18.0) | - | 0.4 | 6.09 (3.33, 8.84)×10⁵ | 5.7 (4.95, 7.85)×10⁵ | - | 1 |
| *C. lusitaniae* | ATCC 34449 | 18.6(17.85, 18.77) | 16.78 (16.53, 16.85) | 23.52 (23.22, 23.82) | 0.1 | 2.6 (1.7, 3.07)×10⁷ | 1.69 (1.69, 1.70)×10⁷ | 5.20 (4.85, 5.55) ×10⁵ | 0.7 |
| *C. parapsilosis* | ATCC 22019 | 25.75 (25.49, 27.34) | 26.72 (26.22, 27.8) | - | 1 | 2.2 (2.0, 3.6)×10⁵ | 1.60 (1.33, 1.68)×10⁶ | - | 0.1 |
| *C. tropicalis* | ATCC 750 | 17.3(17.09, 17.32) | 18.17 (17.74, 18.5) | - | 0.268 | 1.5 (1.25, 1.5)×10⁵ | 4.0 (3.75, 6.25)×10⁵ | - | 0.0765 |
| *C. glabrata* | **Total** | 52.47 (48.42, 56.02) | 19.92 (18.81, 20.48) | 16.4(15.69, 16.77) | 0.000000735 | 3 (1.7, 3.98)×10⁵ | 6.5 (5.0, 10.0)×10⁵ | 9.5 (6.73, 16.75)×10⁵ | 0.000395 |
| | ATCC 2950 | 46.45 (44.55, 46.96) | 17.93 (17.53, 17.94) | 14.28 (13.41, 14.47) | 0.1 | 2.00 (1.35, 2.50)×10⁵ | 6.00 (6.00, 7.80)×10⁵ | 9.00 (7.65, 13.5)×10⁵ | 0.0765 |
| | CL1 | 52.65 (52.49, 53.07) | 18.52 (17.81, 19.22) | 16.50 (16.06, 16.67) | 0.2 | 2.40 (1.40, 2.70)×10⁵ | 3.90 (2.85, 4.95)×10⁵ | 9.00 (7.50, 10.5) ×10⁵ | 0.8 |
| | CL2 | 53.1(53.00, 54.15) | 20.4(19.75, 20.55) | 16.40 (16.40, 16.40) | 0.1 | 3.90 (2.45, 4.70)×10⁵ | 7.00 (6.00, 8.50)×10⁵ | 8.00 (6.50, 9.00)×10⁵ | 0.2 |
| | CL3 | 51.4(51.25, 52.00) | 19.9(19.65, 20.35) | 16.60 (16.00, 16.95) | 0.1 | 4.00 (3.50, 4.00)×10⁵ | 7.00 (5.00, 8.50)×10⁵ | 8.00 (6.50, 14.5)×10⁵ | 0.5 |
| | CL4 | 51.90 (49.71, 53.96) | 20.64 (20.51, 20.77) | 16.17 (16.03, 16.70) | 0.1 | 3.40 (2.15, 4.60)×10⁵ | 7.00 (5.50, 8.50)×10⁵ | 2.10 (1.70, 2.20)×10⁶ | 0.4 |
| | CL5 | 52.90 (50.21, 53.79) | 20.18 (19.93, 20.63) | 17.03 (16.62, 17.35) | 0.1 | 2.00 (1.80, 7.00)×10⁵ | 1.20 (0.85, 1.35)×10⁶ | 1.30 (0.75, 2.00)×10⁶ | 0.268 |

[a] Comparison of Old FA Plus and New FA Plus bottles using Mann-Whitney U test.

[b] ATCC, American Type Culture Collection; NCCP, National Culture Collection for Pathogens of Korea; CL, Clinical isolates.

* Positive only in one bottle among triplicates.

## Retrospective review for clinical blood culture results

In total, 99 isolates of candidemia were analyzed (Fig 1). S1 Table summarizes the number of specimens of candidemia included in two periods. The TTD of *C. glabrata* was significantly shorter in the New FA Plus bottles (45.0 versus 19.4 hours; $P < 0.05$), which was consistent with the results from our horse blood simulation study. The detection time difference between the old and new bottles of *C. glabrata* from peripheral blood (40.6 versus 23.9 hours; $P < 0.05$) was similar to the total blood sample data. For the other three species, overall, the mean TTD was increased in the new bottles. In particular, when analyzing all blood samples for *C. tropicalis* (Old FA Plus 9.58 versus New FA Plus 20.28 hours; $P = 0.0361$) and only peripheral blood samples for *C. parapsilosis* (Old FA Plus 21.33 versus New FA Plus 27.17 hours; $P = 0.02$), New FA Plus showed a statistically significant increase of TTD compared to Old FA Plus.

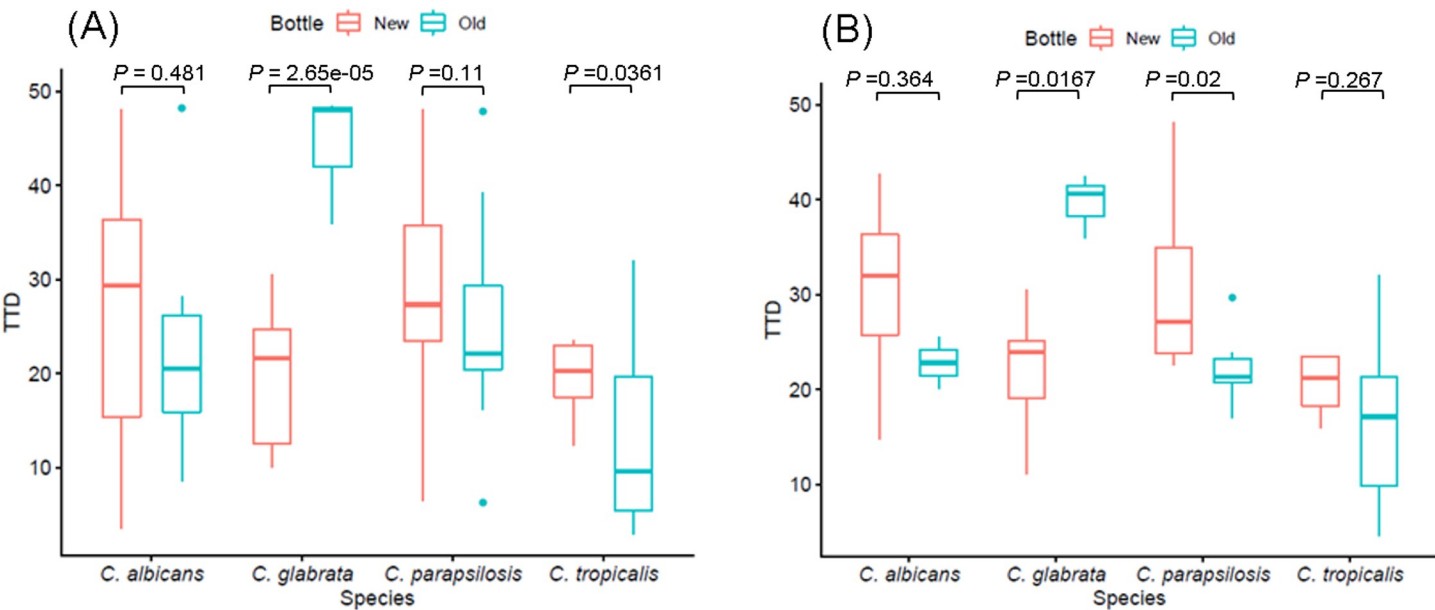

**Fig 1. Time to detection (TTD) of** *Candida* **spp. for the clinical blood culture bottles with different types of aerobic bottles; the New FA Plus (New) and Old FA Plus (Old).** Box plots indicate the medians and the $25^{th}$ and $75^{th}$ percentiles. *P* value indicates the comparison of New and Old bottles using Mann-Whitney U test. (A) All the clinical blood samples were obtained from central line access and peripheral blood. (B) Only peripheral blood samples were included.

## Discussion

Fast and accurate detection of candidemia is crucial in order to choose and start the appropriate antifungal therapy. This study evaluated the TTD and yeast cell counts in positive blood bottles for each *Candida* spp. according to the changes in aerobic bottle media. This study shows that the new aerobic blood culture medium allows for faster detection of *C. glabrata* in the horse blood simulation study and the retrospective clinical sample detection study. The yeast cell counts confirmed that the faster growth of yeast resulted in faster detection of *C. glabrata*.

Altun et al. reported that the median TTD of two *Candida* species was 24.4 hours for *C. albicans* and 54 hours for *C. glabrata* in the Virtuo system using the Old FA Plus bottle with simulated horse blood culture [14]. However, Menchinelli et al. described a marked difference in the TTD; their median TTD of *C. glabrata* in simulated human blood samples was 32.1 and 29.0 hours using the Virtuo and BACTEC (BD Diagnostics) systems and corresponding culture bottles for each system, for the Virtuo system the Old FA Plus bottle was used [15]. The similarity of the results between this study and the Altun et al. study is possibly due to the use of horse blood, which contrasts with the use of human blood in the Menchinelli et al. study. Compared with this study's clinical retrospective data, the median TTD of the Old FA Plus bottle was 45 hours. This result suggests that the human clinical blood culture samples had a shorter detection time in the Virtuo device compared with the simulated data.

The simulation study showed lower median values of TTD for *C. albicans* and *C. guilliermondii* in the New FA Plus than in the Old FA Plus bottle. However, in the retrospective data analysis of the clinical samples, the TTD of *C. albicans* was longer in the New FA Plus bottle, although the difference was not statistically significant. The non-significant difference was because the TTD difference in the simulated study was only slightly different (about 2 hours), and the initial CFU, antibiotic use, and inhibitors in the blood samples and the initial inoculated volume were different according to the clinical conditions of the patients.

Retrospective data analysis results showed that some *Candida* species had an extended TTD due to aerobic culture bottle changes, which can also be related to the simulated study data. In the simulated study, the median TTD values for *C. parapsilosis* and *C. tropicalis* were approximately 1 hour longer in the New FA Plus bottle, but the difference was not statistically significant. In the retrospective analysis, it is difficult to generalize because the TTD prolongation was not significantly observed in all sample groups. However, although the existing blood culture bottle formulation change was known to be substantially equivalent except for the detection of *C. glabrata*, this study suggests the possibility of causing some detection delay for *C. parapsilosis* and *C. tropicalis*.

Although *C. auris* is often known to have multidrug-resistance and several major *C. auris* outbreaks, including bloodstream infections, are reported worldwide, there are few studies on *C. auris* detection using automated blood culture systems, especially the Bact/ALERT Virtuo system [16–19]. This study's simulated blood culture data showed that the *C. auris* detection time using the New FA Plus bottle was delayed about two hours compared with the Old FA Plus bottle, although statistical significance was limited. This result was contrary to the TTD of *C. glabrata*. A recent study conducted a simulation study using sheep blood and reported that the mean TTD of *C. auris* was 29.7 hours in the Old FA Plus Bottle using the Virtuo system [20], which was longer than our data (25.6 hours). The discrepancy in the TTD of *C. auris* in the Old FA Plus bottle between the previous and this study may be due to the differences between sheep and horse blood. Although the difference in the TTD can be regarded as small, further evaluations are needed to determine whether the change in medium ingredients substantially delayed *C. auris* detection, especially in human blood.

The difference in TTD between the simulated and clinical blood culture data may be due to various conditions. In the simulated study, the inoculation concentration of the isolates was 125 CFU/bottle or about 15 CFU/mL; however, the median CFU/mL was 1 CFU/mL in the initial positive candidemia blood culture samples [21]. Also, the differences in plasma components and red blood cell enzymes between horse and human blood may affect fungal growth [22]. However, it has been previously reported that using horse blood does not significantly influence the performance of blood culture systems [14]. A comparison study of horse and human blood has been conducted to demonstrate the equivalence of horse blood for use in blood culture bottle evaluation studies, and it supports that horse blood can be used for blood culture validation studies [22].

This study had several limitations. First, simulated candidemia blood cultures may have led to the differences in growth and detection in *Candida* spp., compared with the clinical specimens. Second, the number of repetitions of the simulation experiment was small, making it difficult to claim any statistical significance. Third, the simulation study only included ATCC strains for the majority of species and clinical isolates were included only for *C. glabrata*. Fourth, in the clinical retrospective blood culture data, only a single institution data was used, and the number of cases was small; therefore, the total number of analyzed cases was insufficient to secure statistical power. In addition, since the clinical retrospective data analysis was conducted only for several *Candida* strains that frequently cause candidemia, it was impossible to confirm whether the delay in TTD of *C. auris* in the New FA Plus bottle during the simulated study was similar to the clinical blood cultures. Fifth, the specific demographic characteristics of the patient group and the blood volume differences among the blood cultures were not considered. The clinical data analysis only included *Candida* spp. positive bottles; therefore, there was a bias in the analysis. Last, in retrospective study, the substantial patient population might be different between two time periods. Although the specimen collections from both periods were obtained from within same institution, patient characteristics may differ between two periods. Furthermore, the number and distribution of isolates included in the

analysis in each period were different, therefore, it is possible that the differences may have affected the results.

Despite these limitations, this study is significant because it showed a reduced TTD compared with previous studies on *C. glabrata* in aerobic bottles. The clinical blood culture results supported the TTD of *C. glabrata* from the simulation study. In addition, our study compared various *Candida* spp., which can result in candidemia, and semi-quantitatively evaluated the growth performance of *Candida* spp. through yeast cell counts, adding to the novelty of this study. In some *Candida* species such as *C. parapsilosis* and *C. tropicalis*, the result that there may be a slight detection delay due to a change in the blood culture bottle is different from what was previously known and seems to require further evaluation.

In conclusion, most *Candida* species did not show remarkable TTD differences, but TTD of *C. glabrata* was markedly reduced in the New FA Plus bottle. This result is meaningful because faster detection and treatment of *C. glabrata*, which is gradually increasing in clinical importance, is possible due to the changes in the media in the blood culture bottle.

## Supporting information

**S1 Table. A summary of the Candida specimen included in retrospective analysis.**
(DOCX)

## Acknowledgments

We thank bioMérieux for providing the BacT/ALERT blood culture bottles used in this study.

## Author Contributions

**Conceptualization:** Mi-Kyung Lee.

**Data curation:** Kye Won Choe.

**Funding acquisition:** Mi-Kyung Lee.

**Investigation:** Kye Won Choe.

**Methodology:** Yong Kwan Lim.

**Supervision:** Yong Kwan Lim, Mi-Kyung Lee.

**Validation:** Yong Kwan Lim, Mi-Kyung Lee.

**Visualization:** Kye Won Choe.

**Writing – original draft:** Kye Won Choe.

**Writing – review & editing:** Yong Kwan Lim, Mi-Kyung Lee.

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
