## [Decision Letter · Decision Letter 0]

29 Aug 2023

PONE-D-23-19815Comparison of New and Old BacT/Alert Aerobic Bottles for Detection of Candida spp.PLOS ONE

Dear Dr. Lee,

Thank you for submitting your manuscript to PLOS ONE. After careful consideration, we feel that it has merit but does not fully meet PLOS ONE’s publication criteria as it currently stands. Therefore, we invite you to submit a revised version of the manuscript that addresses the points raised during the review process.

We look forward to receiving your revised manuscript.

Kind regards,

Ali Amanati

Academic Editor

PLOS ONE

Additional Editor Comments:

Dear authors

Your manuscript [PONE-D-23-19815] has passed the review stage and is ready for ‎revision. ‎

Editorial comments

To ensure the Editor and Reviewers can recommend that your revised manuscript be ‎accepted, ‎‎please pay careful attention to each comment posted underneath ‎this email. This way we ‎can ‎avoid future clarifications and revisions, moving swiftly to ‎a decision.‎

‎1. Please provide a point-by-point response to the Editor and reviewer's comments

‎2. Please highlight all the amends on your manuscript with yellow color

‎3. Improve the English language of the manuscript

Reviewers' comments:

Reviewer's Responses to Questions

**Comments to the Author**

1. Is the manuscript technically sound, and do the data support the conclusions?

Reviewer #1: Yes

Reviewer #2: Partly

Reviewer #3: No

Reviewer #4: Yes

Reviewer #5: Partly

2. Has the statistical analysis been performed appropriately and rigorously? 

Reviewer #1: No

Reviewer #2: Yes

Reviewer #3: Yes

Reviewer #4: No

Reviewer #5: Yes

3. Have the authors made all data underlying the findings in their manuscript fully available?

Reviewer #1: Yes

Reviewer #2: Yes

Reviewer #3: Yes

Reviewer #4: Yes

Reviewer #5: No

4. Is the manuscript presented in an intelligible fashion and written in standard English?

Reviewer #1: Yes

Reviewer #2: No

Reviewer #3: Yes

Reviewer #4: Yes

Reviewer #5: No

5. Review Comments to the Author

Reviewer #1: Choe et al. report on an interesting study concerning a new blood culture bottle from Bioemerieux in the Bact/Alert series, which should speed up the detection of C. glabrata. The study is well presented, that data are appropriately related to the literature, and there is a good discussion and explanation on the weaknesses of the study, e.g., the inoculum used as compared to the clinical consiiton, and the use of horse blood instead of human blood.

I have a couple of comments:

1. The aurhors need to review their statistical analysis of their data. From the CFUs reported in Table 2, it seems that the data may be skewed in direction of Poisson-distibutions - SD is larger than mean in several cases, and this also appears to be true for the TDD in Figure 1, where medians and and percentiles are shown. Here, data in Figure 2, the authors use T-tests for comparisons, which should instead be non-parametric tests such as the Mann-Whitney test. I wonder, whether the same counts for the CFU data in Table 2, CFUs are usually Poisson distributed. Also, since some of the NF-bottles were negative, again it is more correct to use medians and percentiles. It may not change the results but is more correct.

2. The hematocrit data for horse blood should be moved to the Results section.

Reviewer #2: Mi-Kyung Lee presents a laboratory based study comparing the updated BacT/Alert blood culture bottles to the previous version of the bottle.

Abstract: TTD is not written in full in the abstract. This should be done as readers will likely read the abstract prior to the full paper.

Introduction:

- Line 55- 56: what are the changes in the new bottle compared to the old bottle?

Methods:

- Suggest using the new names (based on taxonomic changes e.g. Nakaseomyces glabrata)

- Table 1 - spelling error on "guilliermondii"

- Why were clinical isolates included only for C.glabrata, not the other species

Retrospective data analysis: were there any diagnostic or management differences between the 2 time periods which could have affected the TTD of the 2 bottles. Were the 2 periods collecting specimens from the same patient population?

Results:

- Line 150: how many of each species was included in each period? Overall were there a similar number of isolates in both periods?

- greater variability in TTD as per figure 1. These findings are inconsistent with the conclusion of the paper which suggests that the newer bottle performs better. It only performed better for 1 species.

Line 155: "For the other 3 species...there was no significant difference" This statement is incorrect. For C.parapsilosis, TTD was much longer with the new bottle (p= 0.05)...which is bordering on a significant difference.

Discussion:

- line 181: spelling error again on C.guilliermondii

- Line 188 - Suggest reword to "Although C.auris is resistant to multiple antifungal agents"

- line 189 - blood infections should be bloodstream infections

Additional Limitations:

- only including ATCC strains for the majority of species and including clinical isolates only for C.glabrata

- comparing retrospective clinical data over 2 separate time periods

Reviewer #3: Dear Authors,

In your study, in which you compared the reproduction times of Candida species in your study, it does not include any information or discussion about the blood culture system, except for the time period. In order for your study to have a scientific result, you should reveal the differences in the old and new blood culture bottles, and the issues that can be developed technically or that are recommended to be developed should be expressed. The scientific contribution of the study in this state is very limited. The introduction, material, method and discussion parts of the study contain insufficient or inappropriate information. The work in its current form is unacceptable to me.

Best regards

Reviewer #4: Thanks for your invitation to review the manuscript entitled” Comparison of New and Old BacT/Alert Aerobic Bottles for Detection of Candida spp”. according to my opinion, the number of samples in this study is low, and new and important methods are not used. It is better to publish this manuscript after revision as a report or a short article.

Comments:

- Please don’t use abbreviate words in the Title and Abstract.

- In this sentence” The overall and clinical samples of C. glabrata showed similar results. C. albicans (27.6 versus 25.1 hours; P < 0.001) and C. guilliermondii (28.8 versus 27.3 hours; P = 0.026) also showed significantly shorter TTDs between the Old and New FA Plus bottles”. The P value calculation is not correct because there is no significant difference between the presented numbers.

Reviewer #5: Concerns;

1. For the abstract, please ensure full meanings of words/expressions are written for the first time and then subsequently, abbreviations could be used.

2. Line 53 "It has been reported that C. glabrata was more affected by the blood culture system and

medium than C. albicans" this statement is vague, please clarify, affected in what way.

3. The methods/materials should start by stating clearly what study design this research employed and the study settings. This will help readers to understand the study concept from the start.

4. Without going deep into proprietary issues, the authors should at least describe in basic terms what difference there is among the culture media employed.

5. The conclusion does not bring out the full picture of this study. The absence of significant difference in the other Candida species between the new and old media should at least be acknowledged and then the major finding of interest (i.e C. glabrata) and its importance highlighted.

6. Also, the authors should have explained in the methods or discussion what the new FA Plus bottle differentially has to make this conclusion "This study shows that the TTD of C. glabrata was markedly reduced in the New FA Plus bottle due to the optimized growth performance". Otherwise, "due to the optimized growth performance" could not be concluded from the findings in this study.

6. PLOS authors have the option to publish the peer review history of their article (what does this mean?). If published, this will include your full peer review and any attached files.

Reviewer #1: No

Reviewer #2: No

Reviewer #3: No

Reviewer #4: **Yes: **Parisa Badiee

Reviewer #5: No

---

## [Author Response · Author response to Decision Letter 0]

16 Oct 2023

Responses to the Editor’s and Reviewers’ Comments

Dear reviewers and editorial staffs in Plos One

We are sincerely grateful for your thorough consideration and scrutiny of our manuscript, “Comparison of New and Old BacT/Alert Aerobic Bottles for Detection of Candida spp.”, manuscript No. PONE-D-23-19815R1.

We have revised the manuscript according to the reviewer’s suggestions. We hope that our revised manuscript will be considered and accepted for publication in Plos One. We acknowledge that the scientific and clinical quality of our manuscript was improved by the scrutinizing efforts of the reviewers and editors. The changes within the revised manuscript were highlighted (with yellow color). Point-by-point responses to the reviewers’ comments are provided below. We hope that our responses sufficiently address the reviewer’s concerns.

To. Reviewer #1

Thank you for the valuable insights. We present the responses to your comments below.

Reviewer Point P1.1

The aurhors need to review their statistical analysis of their data. From the CFUs reported in Table 2, it seems that the data may be skewed in direction of Poisson-distibutions - SD is larger than mean in several cases, and this also appears to be true for the TDD in Figure 1, where medians and and percentiles are shown. Here, data in Figure 2, the authors use T-tests for comparisons, which should instead be non-parametric tests such as the Mann-Whitney test. I wonder, whether the same counts for the CFU data in Table 2, CFUs are usually Poisson distributed. Also, since some of the NF-bottles were negative, again it is more correct to use medians and percentiles. It may not change the results but is more correct. 

Reply: We agree with the reviewer on this point about statistical analysis. For statistical analysis of TTD (Time to detection) and CFU data, it may be more appropriate to use the Mann-Whitney U test rather than the t-test because the data is closer to the Poisson-distribution than to the normal distribution. The p-value in Fig 1 was modified to the value using Mann-Whitney U test instead of t test (see Figure R1 below). Also, in Table 2, we performed statistical analysis again using the Mann-Whitney test to determine whether there was a significant difference between new and old aerobic bottles. Additionally, considering the non-parametric distribution of data, the values in table 2 were changed to median, 25 percentile, and 75 percentile values instead of mean and sd. 

Figure R 1 (Fig 1): Time to detection (TTD) of Candida spp. for the clinical blood culture bottles with different types of aerobic bottles; the New FA Plus (New) and Old FA Plus (Old). Box plots indicate the medians and the 25th and 75th percentiles. P value indicates the comparison of New and Old bottles using Mann-Whitney U test. (A) All the clinical blood samples were obtained from central line access and peripheral blood. (B) Only peripheral blood samples were included.

Unfortunately, changes in statistical analysis methods have changed the p-values for several Candida species, making it difficult to claim significant differences. This is thought to be due to the small number of repetitions of the simulation experiment (n=3), and changes in statistical results and interpretation have been added to the results (Line 139-144, 156-158) and discussion (Line 209-210) like below:

 “The overall and clinical samples of C. glabrata showed similar results. C. albicans (27.6 versus 24.95 hours; P =0.2) and C. guilliermondii (28.9 versus 26.9 hours; P = 0.1) also showed relatively shorter TTDs between the Old and New FA Plus bottles, however, the difference was not statistically significant. On the contrary, the TTD was longer for C. auris (25.43 versus 28.25 hours; P =0.0765) in the New FA Plus bottle compared with the Old FA Plus bottle, although statistical significance of the difference could not be clarified.”

 “In addition, although there was no significant difference, median yeast cell counts of New FA Plus was higher than Old FA Plus for C. auris, C. parapsilosis, and C. tropicalis.”

 “The simulation study showed lower median values of TTD for C. albicans and C. guilliermondii in the New FA Plus than in the Old FA Plus bottle.”

Reviewer Point P1.2

The hematocrit data for horse blood should be moved to the Results section.

Reply: We totally agreed with this suggestion. The hematocrit data for horse blood was moved to the Results section (Line 136).

To. Reviewer #2

We greatly appreciate your detailed suggestions to improve the quality of our paper. Responses to each comment are below:

Reviewer Point P2.1

Abstract: TTD is not written in full in the abstract. This should be done as readers will likely read the abstract prior to the full paper.

Reply: We agree with you, the abbreviation in the abstract should have been mentioned in full term beforehand, but was omitted. The full term of TTD (time to detection) has been added to the abstract (Line 24).

Reviewer Point P2.2

Introduction:

- Line 55- 56: what are the changes in the new bottle compared to the old bottle?

Reply: It would have been better if we could have mentioned specific compositional changes as you asked. However, this information could not be obtained because it was proprietary information of the manufacturer, and instead, only a sentence about the approximate information mentioned by the company was added after that sentence (Line 61-63) like below:

“Although the exact formulation of the bottle media is proprietary and not disclosed, it was reported that trace elements were added and several components unrelated to microbial recovery were removed.”

Reviewer Point P2.3

Methods:

- Suggest using the new names (based on taxonomic changes e.g. Nakaseomyces glabrata)

Reply: Thanks to your suggestion, we noticed that based on molecular phylogenetic study, Candida glabrata was recently renamed Nakaseomyces glabratus. However, since this paper mainly focuses on the evaluation of candidemia and we considered that the taxonomy of Candida glabrata has not been completely established, especially in the clinical area, we did not replace all C. glabrata with N. glabratus and added the new name in parentheses after first mentioned C. glabrata (Line 53).

Reviewer Point P2.4

- Table 1 - spelling error on "guilliermondii"

Reply: Thank you for finding the error. The spelling error was corrected.

Reviewer Point P2.5

- Why were clinical isolates included only for C.glabrata, not the other species

Reply: As you commented, more valuable results could have been obtained if we had experimented with other Candida species. However, the number of bottles available during the experiment was limited (especially old bottles), and as we mentioned (Line 63-65), the manufacturer's claim mentioned that the change in bottle formulation shortened the detection time of Candida glabrata as the main finding. In this study, we wanted to know the results of clinically derived isolates of Candida glabrata rather than other species, so clinical isolates of other species were not included in the study.

Reviewer Point P2.6

Retrospective data analysis: were there any diagnostic or management differences between the 2 time periods which could have affected the TTD of the 2 bottles. Were the 2 periods collecting specimens from the same patient population?

Reply: Although the specimen collections from both periods were obtained from patients within the same institution, they cannot be considered the same patient population. We did not intentionally designate or exclude specific patient groups, and samples were included in this study as soon as they were received, so we believe there is no bias, but the results may have been affected. Therefore, the time period-related issue was additionally mentioned in the discussion as a limitation (Line 261-263):

“Last, in retrospective study, the substantial patient population might be different between two time periods. Although the specimen collections from both periods were obtained from within same institution, patient characteristics may differ between two periods.”

Reviewer Point P2.7

Results:

- Line 150: how many of each species was included in each period? Overall were there a similar number of isolates in both periods?

Reply: A summary of the species of specimens included in each period has been added as a supplementary table, S1 table. Since the number and distribution of isolates included in the analysis in each period were different, these differences were added to the discussion (Line 264-266):

“Furthermore, the number and distribution of isolates included in the analysis in each period were different, therefore, it is possible that the differences may have affected the results.”

Table R 1 S1 Table A summary of the Candida specimen included in retrospective analysis

Candida species Old FA Plus

Number of blood specimen

(Peripheral blood specimen) New FA Plus

Number of blood specimen

(Peripheral blood specimen)

Candida albicans 8(2) 19(10)

Candida glabrata 8(3) 11(7)

Candida parapsilosis 16(6) 13(8)

Candida tropicalis 17(11) 7(3)

Total 49(22) 50(28)

Reviewer Point P2.8

- greater variability in TTD as per figure 1. These findings are inconsistent with the conclusion of the paper which suggests that the newer bottle performs better. It only performed better for 1 species.

Line 155: "For the other 3 species...there was no significant difference" This statement is incorrect. For C.parapsilosis, TTD was much longer with the new bottle (p= 0.05)...which is bordering on a significant difference.

Reply: As you commented, this paper claims that Candida glabrata showed better performance, but there was no clear performance improvement in other species. Reviewing Figure 1 (revised using Mann-Whitney U test, not t-test), the TTD of C. tropicalis increased significantly (p=0.0361) in the new bottle, and similarly, the TTD of C. parapsilosis was longer (p=0.02) in the new bottle when only peripheral blood specimens were included (Figure 1B). Therefore, the following was added to the results (Line 179-183) and discussion(Line 216-224) in the main text:

“In particular, when analyzing all blood samples for C. tropicalis (Old FA Plus 9.58 versus New FA Plus 20.28 hours; P =0.0361) and only peripheral blood samples for C. parapsilosis (Old FA Plus 21.33 versus New FA Plus 27.17 hours; P =0.02), New FA Plus showed a statistically significant increase of TTD compared to Old FA Plus.”

“Retrospective data analysis results showed that some Candida species had an extended TTD due to aerobic culture bottle changes, which can also be related to the simulated study data. In the simulated study, the median TTD values for C. parapsilosis and C. tropicalis were approximately 1 hour longer in the New FA Plus bottle, but the difference was not statistically significant. In the retrospective analysis, it is difficult to generalize because the TTD prolongation was not significantly observed in all sample groups. However, although the existing blood culture bottle formulation change was known to be substantially equivalent except for the detection of C. glabrata, this study suggests the possibility of causing some detection delay for C. parapsilosis and C. tropicalis.”

Reviewer Point P2.9

Discussion:

- line 181: spelling error again on C.guilliermondii

- Line 188 - Suggest reword to "Although C.auris is resistant to multiple antifungal agents"

- line 189 - blood infections should be bloodstream infections

Reply: We apologize for repeated spelling errors, and the errors been corrected. 

The sentence for which a reword was requested was modified as follows (Line 225-226): “Although C. auris is often known to have multidrug-resistance”. 

“Blood infections” was replaced to “Bloodstream infections”. Thank you for reviewing the shortcomings of the overall paper.

Reviewer Point P2.10

Additional Limitations:

- only including ATCC strains for the majority of species and including clinical isolates only for C.glabrata

- comparing retrospective clinical data over 2 separate time periods

Reply: We agree with the additional limitations of this study you raised and have added them to the discussion (Line 251-252).:

“Third, the simulation study only included ATCC strains for the majority of species and clinical isolates were included only for C. glabrata.”

To. Reviewer #3

Comment from reviewer:

Dear Authors,

In your study, in which you compared the reproduction times of Candida species in your study, it does not include any information or discussion about the blood culture system, except for the time period. In order for your study to have a scientific result, you should reveal the differences in the old and new blood culture bottles, and the issues that can be developed technically or that are recommended to be developed should be expressed. The scientific contribution of the study in this state is very limited. The introduction, material, method and discussion parts of the study contain insufficient or inappropriate information. The work in its current form is unacceptable to me.

Responses to comment

We accept your critical opinion that this study may have limited scientific contribution. As previously mentioned in Reviewer point P2.2, the details of the bottle composition changes were unknown due to proprietary issues and were not the focus of this study. Rather than exploring the scientific mechanism caused by changes in the composition of the bottle, this study reported and considered changes in detection time of several Candida species that may have clinical implications. In addition, this study focused on the research methodology of a simulation study using horse blood rather than human blood. In response to criticism that this study provided insufficient and inappropriate information, we tried to improve it through this revision, and hope that it has reached an acceptable level for publication.

To. Reviewer #4

Thank you for your thoughtful comments. I understand what you suggested "It is better to publish this manuscript after revision as a report or a short article.", but since this journal only publishes research articles, it seems impossible at this stage. Instead, we have tried to make overall improvements through revision, so we would appreciate it if you could take this into consideration.

Reviewer Point P4.1

- Please don’t use abbreviate words in the Title and Abstract.

Reply: As you commented, we checked all abbreviations of title and abstract and replaced them with full terminology.

Reviewer Point P4.2

- In this sentence” The overall and clinical samples of C. glabrata showed similar results. C. albicans (27.6 versus 25.1 hours; P < 0.001) and C. guilliermondii (28.8 versus 27.3 hours; P = 0.026) also showed significantly shorter TTDs between the Old and New FA Plus bottles”. The P value calculation is not correct because there is no significant difference between the presented numbers.

Reply: We agree with the questions you raised about statistical significance. The t-test was used for the statistics of the manuscript before revision, but as mentioned in Reviewer point P1.1 above, we judged that parametric analysis may not be appropriate because the number of observations is small and the data is not normally distributed. Instead, statistical analysis was performed again using the median value and the Mann-Whitney U test and modified as shown in table 2. Accordingly, the results and discussion were also modified (Line 139-144, 156-158, 209-210).

To reviewer #5

Reviewer Point P5.1

1. For the abstract, please ensure full meanings of words/expressions are written for the first time and then subsequently, abbreviations could be used.

Reply: As mentioned in reviewer point P4.1 above, we checked all abbreviations of title and abstract and replaced them with full terminology. 

2. Line 53 "It has been reported that C. glabrata was more affected by the blood culture system and medium than C. albicans" this statement is vague, please clarify, affected in what way.

Reply: Since the mainly affected factor was the time until detection of Candida species according the cited literature, it was described more clearly (Line 57-58).

3. The methods/materials should start by stating clearly what study design this research employed and the study settings. This will help readers to understand the study concept from the start.

Reply: A summary of the overall study design has been added to the first paragraph of methods/materials (Line 74-79). We hope this will help readers understand the paper.

4. Without going deep into proprietary issues, the authors should at least describe in basic terms what difference there is among the culture media employed.

Reply: As you mentioned, excluding parts that cannot be confirmed due to proprietary issues, we have added the formulation change of the new medium and the composition of the existing medium to the introduction and method (Line 61-63, 96-101).

5. The conclusion does not bring out the full picture of this study. The absence of significant difference in the other Candida species between the new and old media should at least be acknowledged and then the major finding of interest (i.e C. glabrata) and its importance highlighted.

Reply: We have added information about differences in the other Candida species to the conclusion (Line 275-276).:

“In conclusion, most Candida species did not show remarkable TTD differences, but TTD of C. glabrata was markedly reduced in the New FA Plus bottle.”

6. Also, the authors should have explained in the methods or discussion what the new FA Plus bottle differentially has to make this conclusion "This study shows that the TTD of C. glabrata was markedly reduced in the New FA Plus bottle due to the optimized growth performance". Otherwise, "due to the optimized growth performance" could not be concluded from the findings in this study.

Reply: Considering your comment, we performed the yeast cell count study to provide evidence for optimized growth performance, but this alone appears to be insufficient to draw a conclusion. Therefore, we deleted the phrase about optimized growth performance in the conclusion (Line 276).

Again, thank you for giving us the opportunity to strengthen our manuscript with your valuable comments and queries. We have worked hard to incorporate your feedback and hope that these revisions persuade you to accept our submission.

We apologize for the extremely long delay.

Sincerely,

Mi-Kyung Lee, M.D., Ph.D. 

102, Heukseok-Ro, Dongjak-Ku, Seoul 06973, South Korea 

Tel: +82-2-6299-2719, Fax: +82-2-6298-8630

E-mail: cpworld@cau.ac.kr

---

## [Decision Letter · Decision Letter 1]

10 Nov 2023

Comparison of New and Old BacT/ALERT Aerobic Bottles for Detection of Candida Species

PONE-D-23-19815R1

Dear Dr. Mi-Kyung Lee,

We’re pleased to inform you that your manuscript has been judged scientifically suitable for publication and will be formally accepted for publication once it meets all outstanding technical requirements.

Kind regards,

Ali Amanati

Academic Editor

PLOS ONE

Additional Editor Comments (optional):

The current article is scientifically valid in its current form. So, based on my ‎opinion and the respected reviewers' comments could be published.‎

Reviewers' comments:

Reviewer's Responses to Questions

**Comments to the Author**

1. If the authors have adequately addressed your comments raised in a previous round of review and you feel that this manuscript is now acceptable for publication, you may indicate that here to bypass the “Comments to the Author” section, enter your conflict of interest statement in the “Confidential to Editor” section, and submit your "Accept" recommendation.

Reviewer #5: All comments have been addressed

2. Is the manuscript technically sound, and do the data support the conclusions?

Reviewer #5: Yes

3. Has the statistical analysis been performed appropriately and rigorously? 

Reviewer #5: Yes

4. Have the authors made all data underlying the findings in their manuscript fully available?

Reviewer #5: Yes

5. Is the manuscript presented in an intelligible fashion and written in standard English?

Reviewer #5: Yes

6. Review Comments to the Author

Reviewer #5: The authors have managed to use all the available resources and data to re-shape the manuscript in a manner that is more scientifically sound than previously

7. PLOS authors have the option to publish the peer review history of their article (what does this mean?). If published, this will include your full peer review and any attached files.

Reviewer #5: No

---

## [Editor Report · Acceptance letter]

17 Nov 2023

PONE-D-23-19815R1 

Comparison of New and Old BacT/ALERT Aerobic Bottles for Detection of *Candida* Species 

Dear Dr. Lee:

I'm pleased to inform you that your manuscript has been deemed suitable for publication in PLOS ONE. Congratulations! Your manuscript is now with our production department. 

Kind regards, 

on behalf of

Professor Ali Amanati 

Academic Editor

PLOS ONE